# The Comprehensive Approach to Preparation and Investigation of the Eu^3+^ Doped Hydroxyapatite/poly(L-lactide) Nanocomposites: Promising Materials for Theranostics Application

**DOI:** 10.3390/nano9081146

**Published:** 2019-08-10

**Authors:** Katarzyna Szyszka, Sara Targonska, Malgorzata Gazinska, Konrad Szustakiewicz, Rafal J. Wiglusz

**Affiliations:** 1Institute of Low Temperature and Structure Research, Polish Academy of Science, ul. Okolna 2, 50-422 Wroclaw, Poland; 2Wroclaw University of Science and Technology, Polymer Engineering and Technology Division, Wyb. Wyspianskiego 27, 50-370 Wroclaw, Poland; 3Centre for Advanced Materials and Smart Structures, Polish Academy of Sciences, Okolna 2, 50-950 Wroclaw, Poland

**Keywords:** calcium hydroxyapatite nanopowders, rare earth ions, poly(L-lactide), nanocomposites, twin screw extrusion

## Abstract

In response to the need for new materials for theranostics application, the structural and spectroscopic properties of composites designed for medical applications, received in the melt mixing process, were evaluated. A composite based on medical grade poly(L-lactide) (PLLA) and calcium hydroxyapatite (HAp) doped with Eu^3+^ ions was obtained by using a twin screw extruder. Pure calcium Hap, as well as the one doped with Eu^3+^ ions, was prepared using the precipitation method and then used as a filler. XRPD (X-ray Powder Diffraction) and IR (Infrared) spectroscopy were applied to investigate the structural properties of the obtained materials. DSC (Differential Scanning Calorimetry) was used to assess the Eu^3+^ ion content on phase transitions in PLLA. The tensile properties were also investigated. The excitation, emission spectra as well as decay time were measured to determine the spectroscopic properties. The simplified Judd–Ofelt (J-O) theory was applied and a detailed analysis in connection with the observed structural and spectroscopic measurements was made and described.

## 1. Introduction

Regenerative medicine comes in many forms, promising to develop new biomedical treatments for people who suffer due to the burden of trauma, congenital defects and degenerative diseases. Recently, the greatest effort has been put in seeking materials that could help and accelerate the regenerative process by stimulating the body’s own repair mechanisms to functionally heal previously irreparable tissues or organs [1]. Nowadays, the materials with great biological properties, broadly used in medicine, are calcium phosphates and poly(L-lactide) (PLLA) [2].

The biocompatible and biodegradable poly(L-lactide) is one of the most popular polymers produced from renewable raw materials [3]. It can enhance the adhesion and elasticity of composite materials, therefore, it is broadly used in medical and biomedical applications, i.e., for vascular stent production [4], for drug delivery systems [5], for tissue engineering [6], and as surgical sutures, implants and screws. Moreover, it is used in such fields as water purification by oil adsorption [7,8], food packaging [9] or photocatalysis degradation [10]. Poly(L-lactide) can be physically modified by doping it with fillers like bioceramic: β-TCP [11] or chitosan to facilitate cell adhesion, reduce the amount of degradation products, improve cell proliferation, increase hydrophilicity and improve bending between bones and polymeric implants [12]. Among the fillers, phosphates are particularly important, because they are the main inorganic component of vertebrate bones (bone and teeth) [13]. One of the most important materials in regenerative medicine is hydroxyapatite (hereafter: HAp) [14,15,16], which is biocompatible, bioactive and able to form a chemical bond with living tissues. Hydroxyapatite has been broadly investigated as a carrier meant for drug and gene delivery, cellular imaging and biosensing [17,18]. Moreover, it is relatively simple to obtain and modify. A formation of composites based on a PLLA and HAp matrix leads to combining their own advantages and overcoming their disadvantages aimed at bone tissue engineering scaffolds. During degradation of the PLLA/HAp material in the human body, the phosphates are gradually released and transformed into natural bone tissues that can raise the osseointegration of the composite calcium phosphate bioceramics [19].

Composites and nanocomposites based on bioresorbable thermoplastic aliphatic polyesters, i.e., poly(L-lactide) (PLLA), polycaprolactone (PCL) or copolymers thereof (like PGLA) and calcium hydroxyapatites have been widely studied due to their potential applications in regenerative medicine, especially for bone tissue engineering [12,20,21,22,23,24,25,26,27]. Usually, these systems can be used as implants for small bone defects or as scaffolds [28,29].

Many ternary systems based on PLLA/HAp and a third, bioactive component have also been investigated. Many components have been used as antibacterial dopants for PLLA, i.e., chitosan [30,31] or nano silver [32]. Lately, many articles have reported the modification of hydroxyapatite through the introduction of inorganic functional agents like silver [33,34,35,36], zinc [34,36], gold [34,35], copper [35,36] into HAp to achieve antibacterial properties or enhance osteoblast adhesion and proliferation [37]. Furthermore, nanomaterials based on lanthanide compounds are great candidates for biofluorescence probes due to the narrow emission bands, long lifetimes, low photobleaching and relatively low toxicity in comparison with conventional organic dyes. Their long emission lifetimes enable using time-resolved spectroscopy for eliminating the fluorescence of tissue [38,39,40,41]. The last-mentioned properties are especially important and can be used in theranostics for “personalized medicine” with a dual diagnostic and therapeutic function for in vivo imaging applications [42,43].

In this study, we present the synthesis and physico-chemical properties of composites based on bioresorbable PLLA and Eu^3+^ doped hydroxyapatite obtained through twin-screw, co-rotating micro-extrusion as compostes aimed at theranostic application (bone regeneration promotion and bio-imaging possibility).

## 2. Materials and Methods

### 2.1. Materials

Poly(L-lactide) Resomer L210s (PLLA) supplied by Evonik (Darmstadt, Germany) and synthetic hydroxyapatite were used in our research. Calcium hydroxyapatite and Eu^3+^ doped hydroxyapatite were prepared in our lab according to the procedures described below. As starting substrates, Ca(NO_3_)_2_∙4H_2_O (≥99% Acros Organics, Schwerte, Germany), (NH_4_)_2_HPO_4_ (≥99.0% Fluka, Bucharest, Romania), Eu_2_O_3_ (99.99% Alfa Aesar, Karlsruhe, Germany), NH_3_∙H_2_O (99% Avantor Performance Materials Poland S.A., Gliwice, Poland), and HNO_3_ (ultrapure Avantor Performance Materials Poland S.A., Gliwice, Poland) were used.

### 2.2. Synthesis of HAp and Eu^3+^-Doped HAp Powders

Nanocrystalline powders of pure Ca_10_(PO_4_)_6_(OH)_2_ and that activated by Eu^3+^ ions were synthesized by using the precipitation method. The concentration of dopant ions was set to 1 mol%, 3 mol%, 5 mol% Eu^3+^ in a ratio to the overall molar content of calcium cations as the following process. The stoichiometric amount of Eu_2_O_3_ was digested in an excess of HNO_3_ to receive water-soluble europium nitrate, and then the europium nitrate hydrates were re-crystallized three times in order to eliminate of the HNO_3_ excess. Afterwards, the stoichiometric amount of calcium nitrate and europium nitrate was dissolved in deionized water and mixed together. Then the stoichiometric amount of (NH_4_)_2_HPO_4_ was added to the previous mixture, leading to the fast precipitation of the intermediate product. The pH value of the suspension was adjusted to 10 by ammonia. The reaction mixture was heated and stirred for 3 h. Subsequently, the obtained products were washed several times with de-ionized water and dried at 70 °C for 24 h. As the final products, three types of europium-doped hydroxyapatites (1 mol% Eu^3+^:HAp, 3 mol% Eu^3+^:HAp, 5 mol% Eu^3+^:HAp) were obtained. Calcium hydroxyapatite (HAp) was also prepared as a reference sample.

### 2.3. Preparation of the PLLA/Eu^3+^-Doped Hydroxyapatite Composites

PLLA/Eu^3+^:HAp composites having 10 wt.% of the fillers were prepared using the Thermo Scientific Process 11 (Waltham, MA, USA) co-rotating twin-screw micro-extruder (D = 20 mm, L/D = 40) with screw rotation speed of 200 min^−1^ and barrel temperature profile of 200–180 °C (from hopper to die) in a nitrogen atmosphere. The extruder screw geometry was presented in our previous work [44]. The composites were extruded in a one-step process. PLLA- and Eu^3+^-doped HAp were dried in 80 °C under vacuum for 4 h before compounding. After extrusion, the composites were cooled down in the air and pelletized.

The composites were formed into foils through the casting extrusion technique using the Ultra Micro Cast Film extruder (Labtech Engineering, Sweden/Thailand) having flat die with a width of 75 mm, a conical screw with a diameter ranging from 18 to 8 mm (from hopper to die), L/d = 24, temperature of extrusion of 200 °C and screw speed of 100 rpm. Using these parameters, foils with a thickness of ~100 μm were obtained. The obtained materials are summarized in Table 1.

### 2.4. Characterization

Powder X-ray diffraction patterns were measured in the 2θ range of 2–60° by using a Rigaku Ultima IV (Tokyo, Japan) X-ray diffractometer equipped with Ni-filtered Cu Kα_1_ radiation (Kα_1_ = 1.54060 Å, U = 40 kV, I = 30 mA). The experimental XRD patterns were compared with the standards obtained from the Inorganic Crystal Structure Database (ICSD) and analyzed. The degree of crystallinity from XRD patterns was calculated using deconvolution made in Origin 8.0.

High-resolution transmission electron microscopy (HRTEM) images were done by a Philips CM-20 SuperTwin microscope (Eindhoven, The Netherlands), operating at 200 kV. Sample was prepared by dispersing a small amount of specimen in methanol and putting a droplet of the suspension on a copper microscope gird covered with carbon.

The surface morphology and element mapping of the nanocomposites were observed with a scanning electron microscope equipped with energy dispersive spectroscopy FEI Nova NanoSEM 230 (Hillsboro, OR, USA) with an EDS spectrometer (EDAX Genesis XM4) at an acceleration voltage of 18 kV and spot 3.0. Before observation, a layer of graphite was sprayed uniformly over the samples.

IR spectra were acquired using a Thermo Scientific Nicolet iS10 FT-IR Spectrometer (Waltham, MA, USA) equipped with an Automated Beamsplitter exchange system (iS50 ABX containing a DLaTGS KBr detector), a built-in all-reflective diamond ATR module (iS50 ATR), Thermo Scientific Polaris™ and a HeNe laser as an IR radiation source. IR spectra were recorded at 295 K temperature in the 4000–500 cm^−1^ range in KBr pellets with a spectral resolution of 2 cm^−1^.

The emission, excitation spectra, and luminescence kinetics were recorded using an FLS980 fluorescence spectrometer (Edinburgh Instruments, Kirkton Campus, UK). A 450 W xenon lamp was used as an excitation source. The radiation from the lamp was filtered by a 300 mm monochromator equipped with holographic grating (1800 grooves per mm, blaze 250 nm) for the emission and excitation spectra. A microsecond flashlamp (µF2) was used for the measurements of luminescence kinetics, whereas a Hamamatsu R928P photomultiplier was used as a detector. All the emission and excitation spectra were corrected according to the apparatus characteristics and the excitation source intensity. The luminescence kinetics was recorded at 616 nm according to electric dipole transition (^5^D_0_ → ^7^F_2_). The powders were placed in quarc tube and the composites were placed directly in the holder in the spectrometer.

The temperatures of phase transitions were evaluated using the DSC1 STARe Differential Scanning Calorimeter System from Mettler-Toledo (Giessen, Germany). The research was conducted in a nitrogen environment for a temperature range of 25 ÷ 200 °C. The heating/cooling rate was 5 K/min and the gas flow rate remained at 20 mL/min (two thermal cycles).

Samples for mechanical tests were cut out from the obtained foils, from the middle region having ~100 μm, using a 5A type cutter (PN-EN ISO 527-2). Tensile properties tests were conducted on a universal Instron 5966 (Norwood, MA, USA) machine with the speed of 1 mm/min (for Young’s modulus) and 10 mm/min (for tensile strength and strain at break). Before measurements, samples were kept in 80 °C for 2 h to reduce internal stresses.

## 3. Results

### 3.1. Morphology

The morphology of the received hydroxyapatite nanopowders doped with 3 mol% of Eu^3+^ was examined using the TEM and Selected Area Electron Diffraction (SAED) techniques. The received nanoapatite is nanocrystalline, loosely aggregated, and its shape is irregular (see Figure 1). The crystal phase purity of the obtained hydroxyapatite was additionally proved by a SAED analysis. The particles size distribution based on SEM images of pure HAp and 3 mol% Eu^3+^:HAp particles, as well as representative SEM images of powders were presented on Appendix A.

The SEM pictures of the PLLA/HAp composites are shown in Figure 2, Appendix A. All the materials consist of 90 wt.% of PLLA and 10 wt.% of hydroxyapatite. The main difference between the materials is the Eu^3+^ content in the hydroxyapatite. The pictures indicate a random distribution of fillers in all the systems. The hydroxyapatite particle size in all the composites ranges from a few to over one hundred nanometers. The Eu^3+^ addition does not affect the distribution of HAp in the composites. The EDAX picture (Appendix A) also shows the distribution of basic elements (Ca, P) in the PLLA/5 mol% Eu^3+^:HAp. Additionally, the figure presents the distribution of Eu^3+^ in the system.

### 3.2. Structural Analysis of Eu^3+^-Doped Composites

The crystal phase purity of Ca_10_(PO_4_)_6_(OH)_2_ nanocrystals doped with x Eu^3+^ ions (where x = 1, 3, 5 mol%) was checked with the powder XRD technique and was compared with the reference standard of the hexagonal Ca_10_(PO_4_)_6_(OH)_2_ lattice ascribed to the P6_3_/m space group [45] (ICSD-180315) (see Appendix A). The diffraction patterns of the Eu^3+^:Ca_10_(PO_4_)_6_(OH)_2_ nanoparticles embedded into the poly(L-lactide) composite having the function of optically active ions concentration obtained with the extrusion method are presented in Figure 3. The neat poly(L-lactide) crystallized in the orthorhombic α′-polylactide with P2_1_2_1_2_1_ space group. In this case, the degree of crystallinity was approximately 11% (Appendix A). The deconvolution of XRD curves of the hydroxyapatite-doped PLLA, which revealed amorphous PLLA in all the investigated cases, was performed (see Appendix A). Interestingly, mesophase with an amount in the range of 1.9% to 5.9% was also found in PLLA doped with HAp. The highest mesophase content was found in the systems having the highest Eu^3+^ content (3 and 5 mol%).

Infrared spectra were recorded for the entire materials to get a deeper insight into the structure. The IR spectra of HAp consist of typical active vibrational bands of phosphate and hydroxyl groups (see Appendix A).

The IR spectra of the PLLA/HAp composites comprise vibrational bands related to the hydroxyapatite and poly(L-lactide) (see Figure 4). The typical poly(L-lactide) active stretching and banding vibrations of the –CH_3_ and –CH_2_ groups are observed at 2848.3 and 2915.4 cm^−1^, respectively. The most intense peak at 1749.1 cm^−1^ is connected with –C = O stretching vibration. The symmetric and asymmetric stretching of the C-C(=O)-O group was observed at 1452.1 and 1150 cm^−1^. Peaks at this range can be connected with ester groups existing in polylactide and lactide molecules. The 1380.7 cm^−1^ and 1361.0 cm^−1^ peaks were assigned to the scissor vibration δ_s_(–CH_3_) group and the bending vibration of the δ_1_(–CH_3_) group. The vibration of the ester group (–C-O-) derived from poly(L-lactide) molecules can also be clearly observed at 1180.7 cm^−1^. Peaks lying at higher energy corresponded to the vibration of groups belonging to HAp. Regarding the infrared spectra of pure hydroxyapatite, it was possible to determine all of the functional groups of hydroxyapatite. In the case of polylactide composites, the position of phosphate bands is shifted slightly. The triply degenerate ν_3_ antisymmetric stretching of the phosphate groups is observed at 1039.4 cm^−1^ (ν_3_) and 1082.8 cm^−1^ (ν_3_). The ratio intensity of these peaks in reference to nanopowders (Figure 4) is different. In the composite, the peak around 1080 cm^−1^ is more intense, because the ester group vibration (-C-O-) of poly(L-lactide) and the phosphate group vibration (v_3_(PO_4_^3−^)) of hydroxyapatite are overlapped. The symmetric stretching (ν_1_) of the PO_4_^3−^ groups was assigned to the peak at 955.5 cm^−1^. The spectra of all obtained composites show great similarity and are in agreement with the literature data [46,47,48].

### 3.3. Thermal Properties

More information about the supramolecular structure of PLLA in the composites was obtained from DSC measurements. Upon the first heating, the DSC curves of PLLA and the composites are presented in Figure 5a, characteristic phase transitions are visible, such as glass transition with T_g_ at around 61°C, cold crystallization at the mean temperature range of 80 °C ÷ 120 °C, and melting with preceding small exothermic effect corresponding to alpha’-alpha reorganization [49,50]. The thermal parameters estimated form the first heating DSC curves are collected in Appendix A.

The influence of HAp particles and Eu^3+^ ions doping HAp on melt crystallization of PLLA can be analyzed based on the cooling DSC curves presented in Figure 5b and Appendix A. For comparison of the cooling DSC curves, it is clearly visible that melt crystallization exotherms of PLLA in the analyzed samples have different temperature ranges. Neat PLLA and PLLA with unmodified HAp crystallize upon cooling at lower temperature ranges than PLLA with HAp modified with Eu^3+^. The Eu^3+^ ions have an influence on crystallization kinetics of PLLA. In the presence of Eu^3+^, PLLA crystallizes faster. Moreover, it is known that at lower temperatures, it crystallizes into alpha and alpha’ crystals whereas at higher temperatures, PLLA crystallizes in the alpha form [49]. It can be expected that the presence of Eu^3+^ has an influence on the crystalline form of PLLA that forms on melt crystallization. The verification of this assumption can be found in the second heating DSC curves (Figure 5c, Appendix A).

### 3.4. Spectroscopic Properties

The excitation emission spectra of all the obtained materials were measured at room temperature, with a recording emission wavelength at 616 nm, which responds to the maximum of the most intense electric dipole transition (^5^D_0_ → ^7^F_2_). The spectra were corrected to the intensity of the excitation source and normalized to the most intense transition. All spectra contain characteristic sharp lines associated with the intraconfigurational 4f-4f transitions of the Eu^3+^ ions, as well as an intense and broad band related to the ligand-to-metal charge transfer (CT) O^2−^ → Eu^3+^ transition located in the UV region. The excitation emission spectra presented in Appendix A belong to Eu^3+^:HAp nanopowders and in Figure 6 belong to PLLA/Eu^3+^:HAp composites. The 4f orbitals of lanthanide ions are well isolated by the external 5s, 5p and 5d shells and well protected against the influence of the crystal field. Due to this feature, the barycenters of the f-f lanthanide electron transitions are weakly affected by the ligand field, and the position of these peaks remains almost independent of the host lattice structure [51,52]. The narrow peaks observed at 299.2 nm (33 422 cm^−1^) were ascribed to the ^7^F_0_ → ^5^F_(4,3,2,1)_, ^3^P_0_ transitions, at 319.7 nm (31 279 cm^−1^) to ^7^F_0_ → ^5^H_(6,5,4,7,3)_, at 363.4 nm (27 518 cm^−1^) to ^7^F_0_ → ^5^D_4_, ^5^L_8_ at 377.0 nm (26 525 cm^−1^) to ^5^L_8_, ^7^F_0_ → G_2_, ^5^L_7_, ^5^G_3_, at 394.4 nm (25 355 cm^−1^) to ^7^F_0_ → ^5^L_6_, at 416.2 nm (24 027 cm^−1^) to ^7^F_0_ → ^5^D_3_ at 465.9 nm (21 464 cm^−1^) to ^7^F_0_ → ^5^D_2_, as well as at 527.0 nm (18 975 cm^−1^) to ^7^F_0_ → ^5^D_1_. The allowed CT transition is strongly affected by electron-lattice coupling, and the peak position depends on the surrounding symmetry of the ion. In the case of PLLA/HAp composites, the CT maximum is located at 253.5 nm (39 448 cm^−1^). It is well known that the substitution of divalent Ca^2+^ by trivalent Eu^3+^ cations requires the charge compensation mechanism. The two mechanisms are well known for the apatite host lattice and have been described previously in [51,53,54].

The room temperature emission spectra of the entire nanopowders and nanocomposites were measured in the spectral range of 500–750 nm under 394.5 nm excitation as a function of optically active ions concentration. The emission spectra of Eu^3+^:HAp nanopowders are collected in Appendix A and the emission spectra of PLLA/Eu^3+^:HAp composites are collected in Figure 7. The spectra were normalized to the ^5^D_0_ → ^7^F_1_ transition. As can be seen, the emission spectra of Eu^3+^ ions are composed of five bands related to the ^5^D_0_ → ^7^F_0,1,2,3,4_ transitions occurring respectively at ca. 573.5 nm (17 437 cm^−1^), 590.0 nm (16 949 cm^−1^), 616.2 nm (16 228 cm^−1^), 653.2 nm (15 309 cm^−1^) and 700.9 nm (14 267 cm^−1^). The most important transitions in the study of the structural and spectroscopic properties of Eu^3+^ ions are the ^5^D_0_ → ^7^F_0,1,2_ transitions.

The ratio of the integral intensities of the ^5^D_0_ → ^7^F_2_ electric dipole transition to the ^5^D_0_ → ^7^F_1_ magnetic dipole transition is needed to evaluate the asymmetry of the coordination polyhedron of europium(III) ions and the variations in the local point symmetry. This is possible due to the fact that the intensity of the ^5^D_0_ → ^7^F_2_ transition is very sensitive to even small changes in the local environment of Eu^3+^ ions in the crystal field, whereas the intensity of the ^5^D_0_ → ^7^F_1_ transition is nearly independent of those influences. If an Eu^3+^ ion is located in a centrosymmetric site, the only permitted transition is the magnetic one. In the opposite case, the electric dipole transition is dominant. The ratio of the relative emission intensities (*R*) is defined by the equation:(1)R=∫D05→F27∫D05→F17

The higher the ratio between these transitions is, the less centrosymmetric the local environment around Eu^3+^ ions becomes. The impact of the Eu^3+^ ions concentration in powders and composites on the R values are presented in Table 2. It is difficult to observe a straight tendency in the case of both materials. With the influence of the Eu^3+^ ions concentration, the R factor increases, so the Eu^3+^ ions local environment became more distorted, but in the case of 5 mol% Eu^3+^:HAp, it decreased in both powders and in composites. Comparing both materials, the R value was higher in composites, which indicates that in these materials, the surroundings of the Eu^3+^ ions is more distorted.

The simplified Judd–Ofelt approach was implemented to provide the intensity parameters Ω_2_ and Ω_4_, as well as to have a deeper insight into the structure. The results of this approach are shown in Table 2. The value of the Ω_2_ parameter indicates some changes in the distortion of europium coordination polyhedra caused by such factors as ions concentration, annealing temperature, etc., and could be related to an increase of the Eu^3+^ − O^2−^ bond covalency. It is worth noting that the value of the Ω_2_ parameter has the same tendency as the R factor in powders and composites. The value of the Ω_4_ factor supplies some information about changes in the electron density around Eu^3+^ cations. This value cannot be straightly correlated with changes in the Eu^3+^ ions symmetry, but it can add some information about the electron density variations of the O^2−^ anions surrounding that influences the CT band position.

To determine the detailed spectroscopic properties of nanopowders and nanocomposites, the luminescence kinetics was analyzed. The luminescence life times were recorded at room temperature. The materials were excited by a 394.5 nm line and monitored at 618 nm corresponding to the ^5^D_0_ → ^7^F_2_ transition. The luminescence kinetic curves of the Eu^3+^:HAp nanopowders are shown in Appendix A, and for the Eu^3+^:Ca_10_(PO_4_)_6_(OH)_2_ nanoparticles embedded into poly(L-lactide) composites, they are shown in Figure 8.

### 3.5. Tensile Properties

In Figure 9, we presented the tensile results for PLLA-based composites with 10 wt.% of HAp with different Eu^3+^ ions concentration. The parameters are very important factors in designing 3D scaffolds for bone tissue engineering. From among three basic methods (polymerization in situ, solution-casting and melt mixing), we chose melt mixing-extrusion in a twin-screw extruder. This method ensures good filler distribution in the polymer. The preparation method is the most important in the case of nanocomposites preparation.

## 4. Discussion

### 4.1. Thermal Properties

The strongest differences in thermal properties of PLLA and the composites concern cold crystallization. In the presence of HAp particles, the onset of cold crystallization (T_cc_^onset^), as well as that of peak temperature (T_cc_), is about 3 °C lower compared to neat PLLA. This result indicates the well-known nucleating activity of HAp particles on PLLA cold crystallization [50]. It is worth underlining that the stronger effectiveness of nucleation activity towards cold crystallization is exhibited by HAp particles doped with Eu^3+^ ions. The lowest T_cc_^onset^ was registered for the PLLA/3 mol% Eu^3+^:HAp composite, the cold crystallization onset is about 9.4 °C lower than for neat PLLA.

Moreover, the enthalpy of cold crystallization (ΔH_cc_) is lower in the case of the PLLA/Eu^3+^:HAp composite than for neat PLLA and PLLA with the unmodified HAp particles composite. The lower values of cold crystallization enthalpy prove the higher crystallinity of PLLA in the presence of Eu^3+^:HAp. The presence of Eu^3+^ ions and its molar content have a strong influence on the crystallinity degree of PLLA in composites.

The lack of exothermic effects of the alpha’-alpha transition in the second heating DSC curves in the case of composites with Eu^3+^ confirms that, upon subsequent cooling, PLLA crystallized in the alpha form. By contrast, neat PLLA and PLLA with unmodified HAp crystallized upon cooling as a mixture of alpha and alpha’ crystals, which is designated by the presence of an exothermic peak at 165 °C corresponding to the alpha’-alpha reorganization.

The PLLA crystallizes faster and in alpha and alpha’ form in the presence of Eu^3+^ ions embedded in the apatite. The HAp doped with Eu^3+^ ions shown high nucleation activity towards cold crystallization as well as high crystallinity degree of PLLA in composites.

### 4.2. Spectroscopic Properties

The red emission was observed from the Eu^3+^:HAp powders and the PLLA/Eu^3+^:HAp composites. The analysis of the ^5^D_0_ → ^7^F_0_ transition can provide data about the number of crystallographic sites substituted by Eu^3+^ ions into the host structure. Additionally, the existence of this transition approves that the Eu^3+^ ions are located at a low-symmetry environment, and it is observed only if the Eu^3+^ ions occupy sites with a local symmetry of C_n_, C_nv_ or C_s_. The ^5^D_0_ → ^7^F_0_ transition is split into three components located at 573.5 nm (17 437 cm^−1^) and 577.2 nm (17 325 cm^−1^) and 578.6 nm (17 283 cm^−1^) in the case of Eu^3+^:HAp nanopowders. The Eu^3+^ ions occupy three different types of crystallographic sites, which is well known in the literature on apatite host matrixes [51,52,53,54]. The ^5^D_0_ → ^7^F_1_ and ^5^D_0_ → ^7^F_2_ transitions consist of many overlapped Stark components. The most intense emission was observed for the hypersensitive ^5^D_0_ → ^7^F_2_ electronic transition. Its intensity is highly influenced by the local symmetry of the Eu^3+^ ions as well as the type of ligands rather than the intensities of the other electronic transitions. This transition is notably used to calculate the asymmetry of the Eu^3+^ site. In the case of europium(III) emission in the PLLA/1 mol% Eu^3+^:HAp composite, it is visible that the ^5^D_0_ → ^7^F_0_ transition is split into three components, but in the case of higher concentration, it is possible to distinguish only a single line. Surprisingly, the major difference is visible in the intensity of the ^5^D_0_ → ^7^F_4_ transition which is much more intense in the case of the composite in the entire range of optically active ions concentration.

The decay profiles are non-exponential. Since the physical meaning of multi-exponential fitting is complicated to explain, the lifetime values were calculated from the effective emission decay times using the following equation:(2)$$\tau_{m}=\frac{\int_{0}^{\infty }tI(t)dt}{\int_{0}^{\infty }I(t)dt}≅\frac{\int_{0}^{t^{\max}}tI(t)dt}{\int_{0}^{t^{\max}}I(t)dt}$$
where *I*(*t*) is the luminescence intensity at time t corrected for the background, and the integrals are calculated over the range of 0 < *t* < *t^max^*, where *t^max^* >> *τ_m_*. The value of decay time is shorter in the case of the Eu^3+^ ions-doped hydroxyapatite embedded into poly(L-lactide). This observation could be connected with different response of the Eu^3+^ ions present on the surface in nanopowders than in composites. The decay times shortened with an increase of Eu^3+^ ions concentration in HAp embedded into PLLA.

### 4.3. Tensile Properties

The tensile strength of the PLLA/HAp composite is ~40.1 MPa, which is comparable to all the three PLLA/Eu^3+^:HAp composites (40.4–44.3 MPa with a standard deviation 2.4–2.9 MPa). In this case, no influence of Eu^3+^ ions on the tensile strength of the composites was observed. However, tensile strength of all the composites is lower compared to neat PLLA. The effect of mechanical properties reduction was described earlier [55] for PLLA doped with cellulose nanocrystals or carbon nanotubes but also for hydroxyapatite [16]. The reason for this effect is polymer–filler adhesion/interaction. In the research, neat calcium hydroxyapatite without any surface modification has been used. In this case, adhesion is rather poor and is not improved by the Eu^3+^ ions incorporation. Another reason is the filler content. It has been shown before that, in the extruded composites, the higher the content of hydroxyapatite, the lower the tensile strength [16]. Young’s modulus all the values for the composites are in the range of 3.1–3.2 GPa with a standard deviation of 0.1–0.3 GPa. When comparing the strain at break parameter, the reduction for PLLA/HAp (2.4%) as compared to neat PLLA (6.7%) can be observed along with further reduction of PLLA/Eu^3+^:HAp (1.4–1.7%) composites as compared to the PLLA/HAp system. The last effect is very slight (0.7–1.0%); however, the measurement error ranges from 0.1% to 0.2%. The tensile strength, Young’s modulus and strain at break parameters value are the highest in the case of neat PLLA and are comparable to each other for all PLLA/HAp composites.

## 5. Conclusions

In the present research, pure as well as Eu^3+^-doped hydroxyapatite were synthesized by using the precipitation method and were confirmed by the XRD patterns analysis. The poly(L-lactide)/europium(III)-doped hydroxyapatite composite foils were successfully fabricated using the twin-screw co-rotating micro-extrusion technique in the weight ratio 9:1. The obtained composites were found to have similar properties to neat PLLA. Differential Scanning Calorimetry revealed some differences in thermal properties of PLLA and the composites. In the presence of HAp particles, the cold crystallization is lower in comparison with neat PLLA. This result indicated a well-known nucleating activity of HAp particles towards the cold crystallization. Moreover, stronger effectiveness of nucleation activity towards cold crystallization was shown in the case of HAp particles doped with Eu^3+^ ions. The HAp nanopowders and composites doped with Eu^3+^ ions are characterized by red emission under UV radiation with the *^5^D_0_ → ^7^F_2_* transition as the most intense. The luminescence kinetics shows a non-exponential curve indicating that more than one emitting center is present in the investigated materials. The value of luminescence decays is lower in the case of the Eu^3+^ ions-doped hydroxyapatite embedded into poly(L-lactide).

The PLLA/Eu^3+^:HAp composites were obtained as prospective candidates to theranostic applications (therapy and diagnostics) due to support of bone healing by hydroxyapatite and bio-imaging possibility of Eu^3+^ ions.

## Figures and Tables

**Figure 1 nanomaterials-09-01146-f001:**
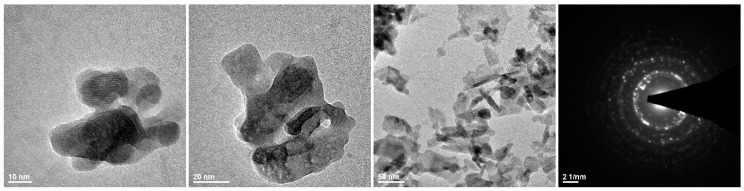
Representative TEM images and a SAED image of the 3 mol% Eu^3+^:Ca_10_(PO_4_)_6_(OH)_2_.

**Figure 2 nanomaterials-09-01146-f002:**
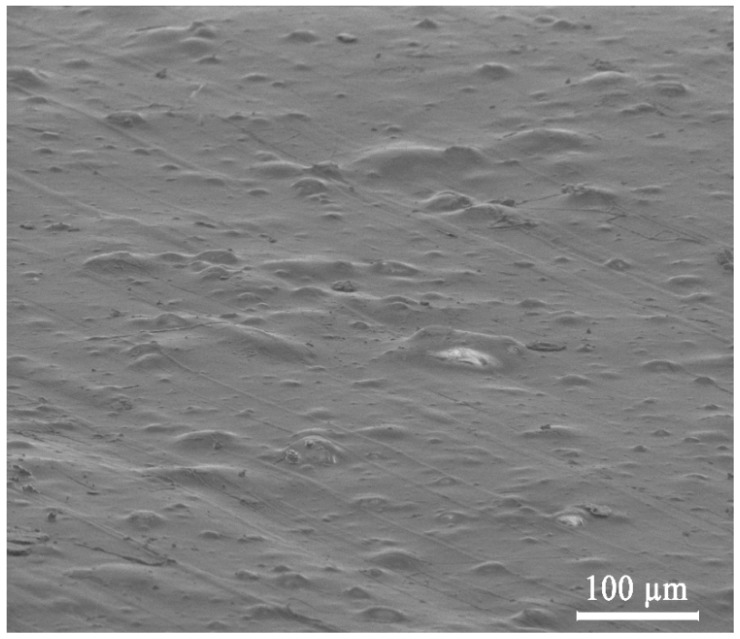
SEM image of composite surface for PLLA/HAp.

**Figure 3 nanomaterials-09-01146-f003:**
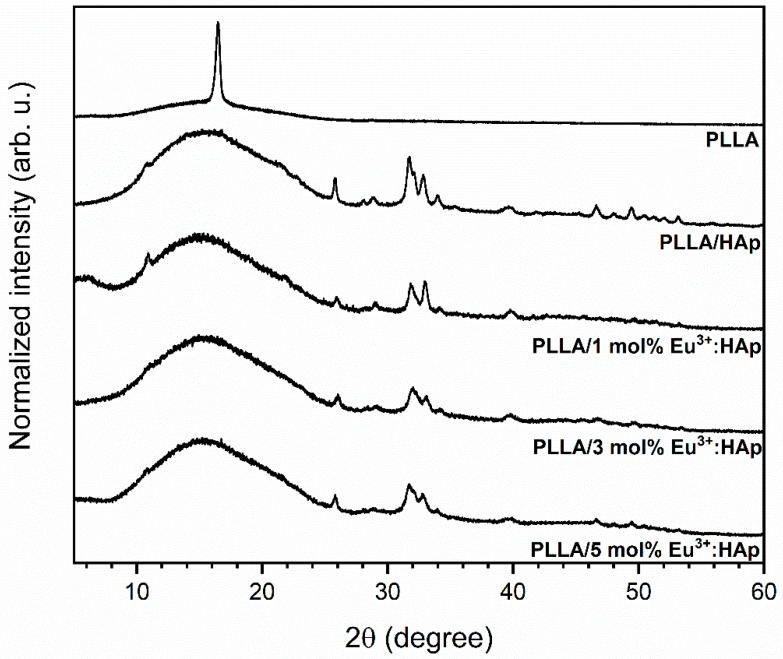
X-ray diffraction patterns of PLLA/x mol% Eu^3+^:HAp composites (where x = 0–5) obtained with the extrusion method.

**Figure 4 nanomaterials-09-01146-f004:**
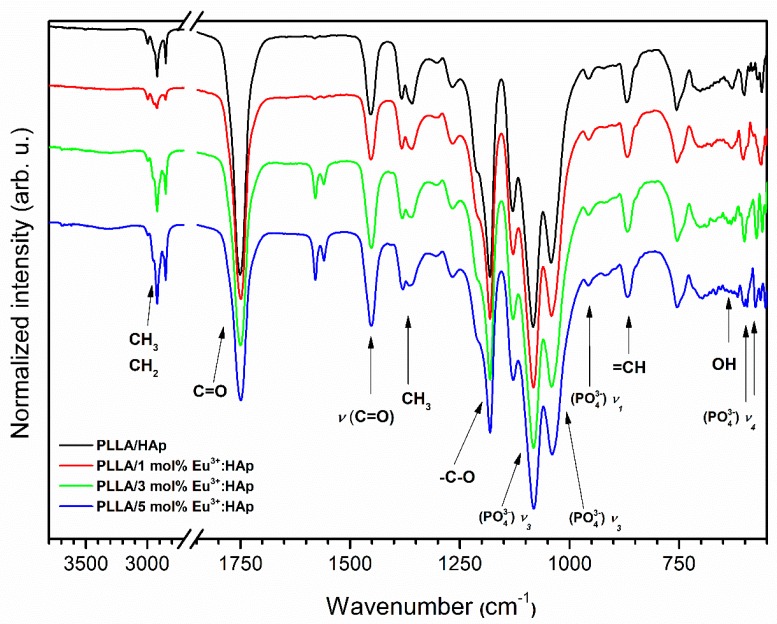
IR spectra of PLLA/x mol% Eu^3+^:HAp composites (where x = 0–5) obtained via extrusion in situ.

**Figure 5 nanomaterials-09-01146-f005:**
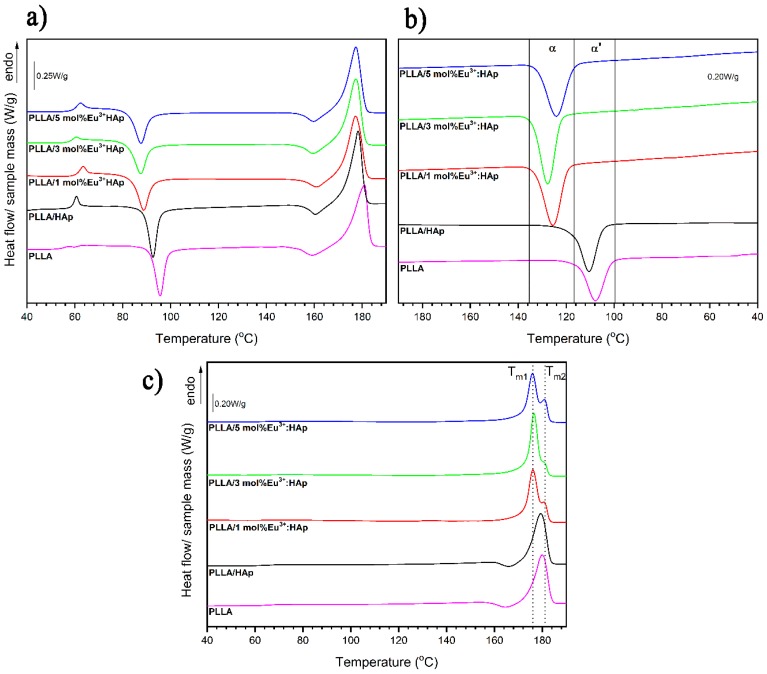
The first heating (**a**), cooling (**b**) and second heating (**c**) DSC curves of PLLA and PLLA/x mol% Eu^3+^:HAp composites (where x = 0–5).

**Figure 6 nanomaterials-09-01146-f006:**
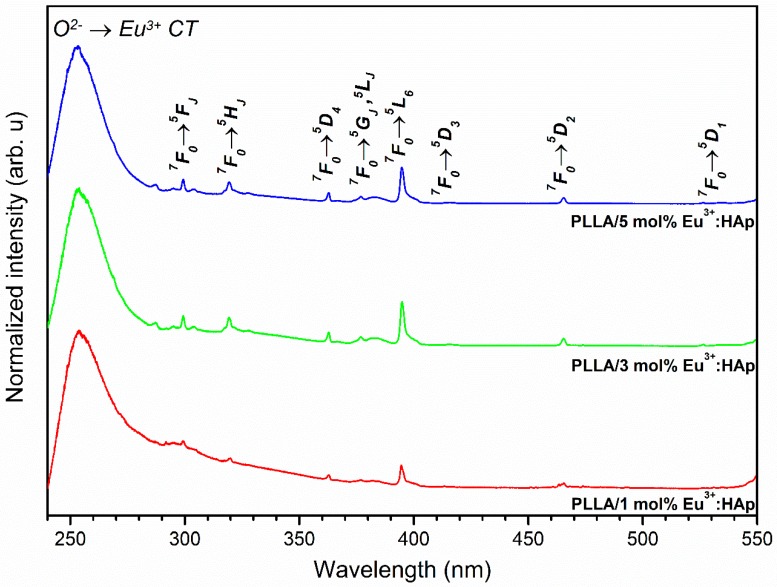
Excitation spectra of x mol% Eu^3+^:HAp nanoparticles (where x = 1–5) incorporated into PLLA composites.

**Figure 7 nanomaterials-09-01146-f007:**
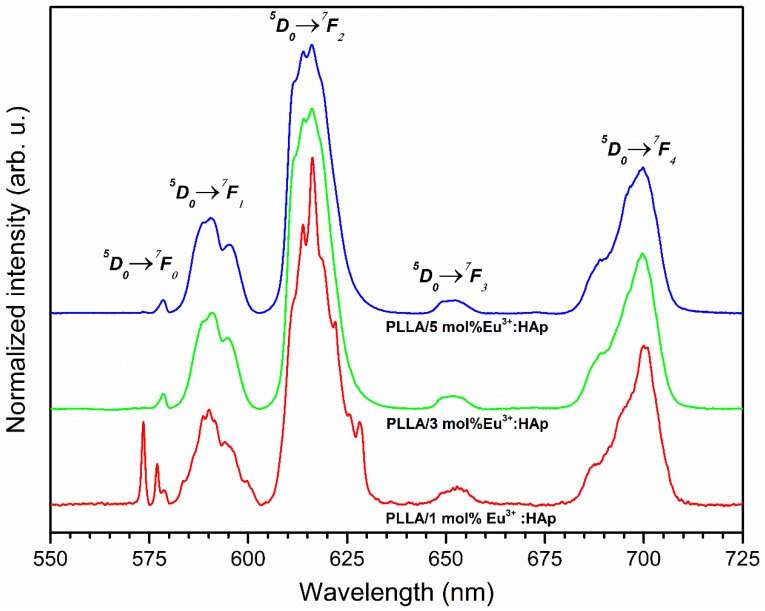
Emission spectra of PLLA/x mol% Eu^3+^:HAp composites (where x = 0–5) incorporated into poly(L-lactide) composites obtained by using the extrusion method.

**Figure 8 nanomaterials-09-01146-f008:**
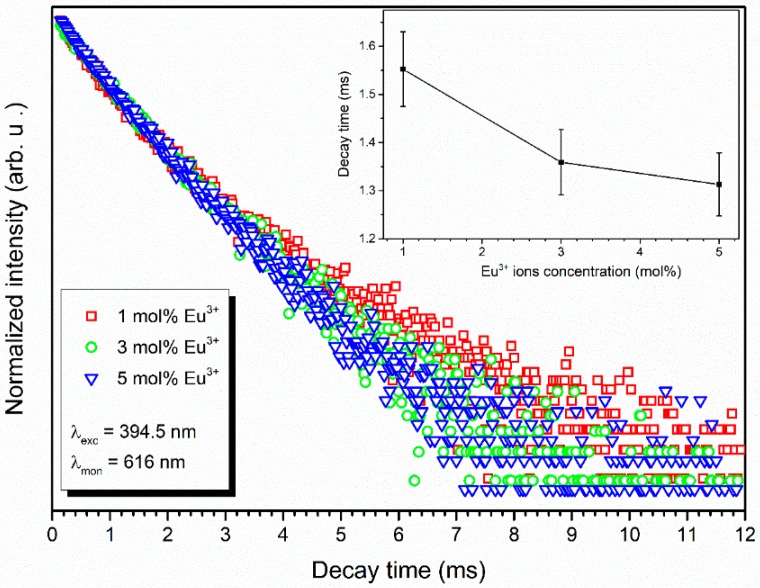
Emission kinetics of the Eu^3+^:Ca_10_(PO_4_)_6_(OH)_2_ nanoparticles embedded into poly(L-lactide) composites obtained by using the extrusion method. Insert: The decay time as a function of Eu^3+^ ions concentration.

**Figure 9 nanomaterials-09-01146-f009:**
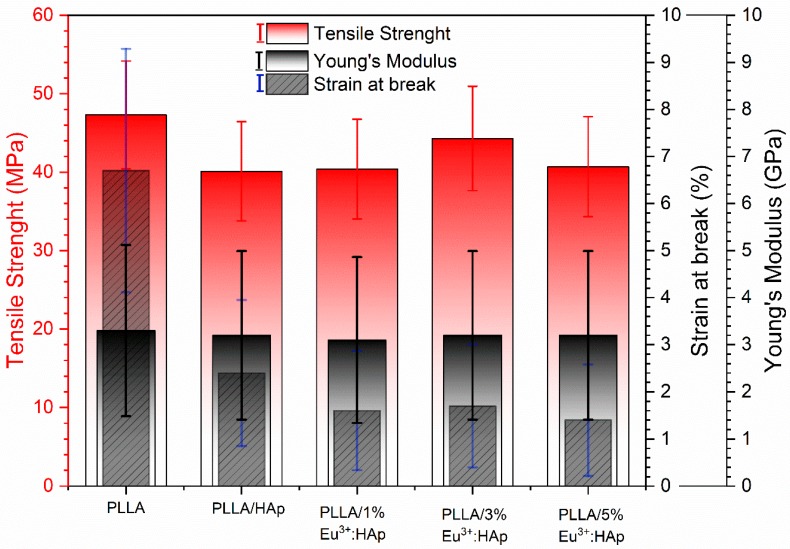
Tensile properties for PLLA/HAp composites: tensile strength (red left axis), strain at break (right axis) and Young’s modulus (right axis).

**Table 1 nanomaterials-09-01146-t001:** Summary of the investigated materials.

Symbol	PLLA (wt.%)	Hap (wt.%)
HAp	0	100
1 mol% Eu^3+^:HAp	0	100
3 mol% Eu^3+^:HAp	0	100
5 mol% Eu^3+^:HAp	0	100
PLLA	100	0
PLLA/HAp	90	10
PLLA/1 mol% Eu^3+^:HAp	90	10
PLLA/3 mol% Eu^3+^:HAp	90	10
PLLA/5 mol% Eu^3+^:HAp	90	10

**Table 2 nanomaterials-09-01146-t002:** Decay rates of radiative (A_rad_), non-radiative (A_nrad_) and total (A_tot_) processes of ^5^D_0_ → ^7^F_J_ transitions, luminescence lifetimes (τ), intensity parameters (Ω_2_, Ω_4_), quantum efficiency (η) and asymmetry ratio (R) of the powders and composites.

**x mol% Eu^3+^:HAp Powders**								
**Sample**	**A_rad_ (s^−1^)**	**A_nrad_ (s^−1^)**	**A_tot_ (s^−1^)**	**T (ms)**	**Ω_2_ (10^−20^ cm^2^)**	**Ω_4_ (10^−20^ cm^2^)**	**η (%)**	**R**
1 mol% Eu^3+^	160.05	305.07	465.12	2.15	4.0441	1.0793	34.41	2.874
3 mol% Eu^3+^	179.94	391.49	571.43	1.75	4.6633	1.3196	31.49	3.314
5 mol% Eu^3+^	155.55	387.93	543.48	1.84	3.7878	1.2680	28.62	2.692
**PLLA/x mol% Eu^3+^:HAp Composites**								
1 mol% Eu^3+^	230.30	414.86	645.16	1.55	4.9835	4.5359	35.70	3.542
3 mol% Eu^3+^	218.28	517.02	735.29	1.36	4.5349	4.5455	29.69	3.223
5 mol% Eu^3+^	201.85	561.50	763.36	1.31	4.0362	4.3208	26.44	2.868

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
