# Peer review of "The Comprehensive Approach to Preparation and Investigation of the Eu3+ Doped Hydroxyapatite/poly(L-lactide) Nanocomposites: Promising Materials for Theranostics Application"

_nanomaterials, 2019, doi:10.3390/nano9081146_

Round 1

Reviewer 1 Report

The manuscript "The comprehensive approach to preparation and investigation of the Eu3+ doped hydroxyapatite/poly(L-lactide) nanocomposites: promising materials for theranostics application" by Wiglusz and coworkers deals with the fomation of PLLA/apatite composite materials. Overall, the manuscript has some interesting indications but it should be elaborated further. There are several comments that should be adressed in a revised version. Also the English style and grammar should be checked again (see examples below). 

General comments:

1. The authors state that the hydroxyapatite particles size ranges from few to over hundered micrometers. Some statistics should be added for that point.

2. How is the HAp distributed in the PLLA? Is it homogeneous?

3. How stable are the composites against water, acid and base? What about the degradability?

4. I think the discussion part should be elaborated further. The results should be discussed more thoroughly. Especially, some explanations why the properties change with the different compositions should be given.

5. Possible applications shoulöd be mentioned and the results discussed regarding these applications. How and why is the described material better than the state of the art?

6. How is the biocompatibility of the material? It should be discussed as well.

7. Why is the error of the PLLA sample in Figure 9 so high?

Specific comments:

Introduction: style "broadly using", "row materials", "important because are", "gene transaction", "leads to combine", "the lanthanide compounds"

Materials: style "to get rid", "70oC", "Composite were formed"

Discussion: "It's worth", "difference it"

Conclusions: "proofed that there has been", "by precipitation way", "the successful", "the XRD", "is presented in"

Author Response

Dear Editor, 

We would like to express our sincerest gratitude to the Reviewers for their efforts put in criticizing the manuscript. We have taken into account all raised question here follows the detailed answers to the Reviewers. Moreover, all changes we have made to the original manuscript, are marked in the red color in the text. 

Comments and Suggestions for Authors

The manuscript "The comprehensive approach to preparation and investigation of the Eu3+ doped hydroxyapatite/poly(L-lactide) nanocomposites: promising materials for theranostics application" by Wiglusz and coworkers deals with the fomation of PLLA/apatite composite materials. Overall, the manuscript has some interesting indications but it should be elaborated further. There are several comments that should be adressed in a revised version. Also the English style and grammar should be checked again (see examples below). 

Answer:

The manuscript has been proofread.

General comments:

The authors state that the hydroxyapatite particles size ranges from few to over hundered micrometers. Some statistics should be added for that point.

Answer:

Figure 1. Particle size distribution based on SEM images (a) as well as SEM images of the pure HAp (b) and 3 mol% Eu3+:HAp.

The particular particles length of the pure as well as the 3 mol% Eu3+-doped hydroxyapatite materials were measured based on SEM images. The results are presented on Figure 1 (above) as a histogram of particle size distribution. Representative SEM images of HAp and 3 mol% Eu3+:HAp powders are shown on Figure 1 b and c, respectively.  The Eu3+-doped crystals are smaller than the pure hydroxyapatite crystals. The mean sizes are 100 nm and 150 nm in length respectively.

How is the HAp distributed in the PLLA? Is it homogeneous?

Answer:

Yes, hydroxyapatite particles are distributed relatively homogeneously in PLLA based on elements mapping signed as Figure S3 in supplementary file.

How stable are the composites against water, acid and base? What about the degradability?

Answer:

The stability and degradability of composites are not the subject of this paper. These parameters are well-known for PLLA, HAp and in the case of composites, these parameters should be very similar.

I think the discussion part should be elaborated further. The results should be discussed more thoroughly. Especially, some explanations why the properties change with the different compositions should be given.

Answer:

These explanations have been added into the manuscript.

Possible applications shoulöd be mentioned and the results discussed regarding these applications. How and why is the described material better than the state of the art?

Answer:

The Eu3+-doped nanohydroxyapatite embedded into PLLA matrix possess additional function as possibility of diagnosis at the use of Eu3+ ions. We obtained composites aimed to theranostic application which possess such advantages as biocompatibility (PLLA, HAp), bioactivity, bone regeneration promotion (HAP) as well as bio-imaging possibility (Eu3+ions).

How is the biocompatibility of the material? It should be discussed as well.

Answer:

The hydroxyapatite as well as poly(L-lacide) are biocompatible and this feature is well-known.
However, we investigated the influence of composites on human adipose-derived stromal cells (hASCs) which was the subject of another paper (Mater Sci Eng C 98 (2019) 213–226).

Why is the error of the PLLA sample in Figure 9 so high?

Answer:

In the paper there has been presented mechanical tests results for PLLA (Poly-L-lactide) composites based on the hydroxyapatites obtained by twin screw extrusion process. As far as it has been known, there is no flawless method of obtaining composite materials. Among three basic methods (polymerization in situ, solution-casting and melt mixing) there has been chosen melt mixing – extrusion in the twin screw extruder. This method has huge advantage providing a good filler (e.g. hydroxyapatite) distribution in polymer matrix. In our case, this is the most important aspect because there have been investigated the nanocomposites. There are another advantages related to this method like rapidity and continuity ect. On the other hand it should be taken into account that the PLLA is very sensitive material for degradation which occurs during extrusion process. In this case, the molecular weight of PLLA decreases and reduces the mechanical properties, especially tensile strength. It could be assumed that it is one of the most important reason. The effect of mechanical properties reduction is not new – it has been described earlier [1] for PLLA doped with carbon nanotube. In our case, taking into account the possible application for bone tissue engineering there has been decided to investigate and show the mechanical tests results to be fair and show also disadvantage of the method and material. Next reason for this effect is related to an adhesion or an interaction of polymer and filler. There has been used the calcium hydroxyapatite without any surface modification to avoid additional impurities, where adhesion is rather poor. It is well known that there are many adhesion promoters or coupling agents that would help in this specific case – we are also working in this area and hopefully we will be able to publish our results. Another reason is a universal Instron 5966 machine that was used to obtain the mechanical properties generating so high errors.  

Specific comments:

Introduction: style "broadly using", "row materials", "important because are", "gene transaction", "leads to combine", "the lanthanide compounds"

Materials: style "to get rid", "70oC", "Composite were formed"

Discussion: "It's worth", "difference it"

Conclusions: "proofed that there has been", "by precipitation way", "the successful", "the XRD", "is presented in"

Answer:

The detailed phrases have been corrected. Moreover, the manuscript has been proofread. 

Bibliography

[1]          I. Armentano, D. Puglia, F. Luzi, C.R. Arciola, F. Morena, S. Martino, L. Torre, Nanocomposites based on biodegradable polymers, Materials (Basel). 11 (2018). doi:10.3390/ma11050795.

Reviewer 2 Report

The manuscript by Wiglusz and co-workers details the preparation and characterization of europium-based nanocomposites for theranostics. The manuscript presents a good amount of data, and the reproducibility was demonstrated and error are shown. It is sufficiently detailed to be reproducible. The topic is of interest to the journal’s readers. However, there are several issues that must be addressed prior to further consideration.

1. Table 1 is confusing as the percentages do not add up to 100% due to the different units used in the table. Either use wt% or mol% for all the ingredients so that the components all add up to 100%. Only this way their relative ratio can be interpreted from the table.

2. The SAED image as an inset in Figure 1 is not legible and cannot be interpreted. In particular, the scale bar. Figure 1 should be presented as 4 equally sized panels in a single row.

3. With regards to the intended applications, the antimicrobial performance of the materials should be discussed either based on similar literature or preferably experimental data for the new materials should be provided. The manuscript in its current form mainly focuses on standard materials characterization only.

4. All the materials, chemicals and solvents along with their purity/grade and supplier should be listed under section 2.1 and not scattered in the manuscript.

5. Various symbols were used for the degree sign in degC. Use the correct symbol throughout the manuscript.

6. Section 2.4 describes the characterization of the materials. The authors should add the details for sample preparation for all methods described in this section.

7. The scope of introduction should be widen to appeal for a broader audience, for instance the versatile applicability of PLLA should be briefly demonstrated in a sentence by mentioning and giving examples for diverse fields, e.g. membrane supports (ACS Sustainable Chem. Eng., 2019, 7, 11885), oil adsorption (Appl. Sci. 2019, 9, 1014), food packaging (Materials, 2017, 10, 952), photocatalysis degradation (ACS Sustainable Chem. Eng., 2018, 6, 2445).

8. The SEM images show the same features for all the materials shown in Figure 2. Consequently, only one representative SEM image will suffice in the manuscript. The rest can be moved to the supporting information.

9. In the caption of Figure 5 the panel designations should not be capital letters in order to be consistent with the rest of the manuscript.

10. The 3 panels of Figure 9 can be merged into a single panel with 3 y-axis.

11. The strain at break for PLLA has a significant error. The authors should elaborate on this and further explanation should be provided.

12. The “Discussion” section is numbered as section #1 but it should be #4 (line 306).

13. Overall it is unclear to the reader what the actual novelty and achievements are. The authors should stress what has already been known in the literature and what is the advancement of the field that was achieved in this work.

14. The conclusion section should have some quantitative statements summarizing the most important research outcomes of the manuscript.

Author Response

Dear Editor, 

We would like to express our sincerest gratitude to the Reviewers for their efforts put in criticizing the manuscript. We have taken into account all raised question here follows the detailed answers to the Reviewers. Moreover, all changes we have made to the original manuscript, are marked in the red color in the text. 

Comments and Suggestions for Authors

The manuscript by Wiglusz and co-workers details the preparation and characterization of europium-based nanocomposites for theranostics. The manuscript presents a good amount of data, and the reproducibility was demonstrated and error are shown. It is sufficiently detailed to be reproducible. The topic is of interest to the journal’s readers. However, there are several issues that must be addressed prior to further consideration.

Table 1 is confusing as the percentages do not add up to 100% due to the different units used in the table. Either use wt% or mol% for all the ingredients so that the components all add up to 100%. Only this way their relative ratio can be interpreted from the table.

Answer:

The last column of Table 1 has been removed for better clarity.

The SAED image as an inset in Figure 1 is not legible and cannot be interpreted. In particular, the scale bar. Figure 1 should be presented as 4 equally sized panels in a single row.

Answer:

The Figure 1 has been changed according to reviewer comment.

With regards to the intended applications, the antimicrobial performance of the materials should be discussed either based on similar literature or preferably experimental data for the new materials should be provided. The manuscript in its current form mainly focuses on standard materials characterization only.

Answer:

The antimicrobial properties of obtained composites are not the subject of presented paper. However, antimicrobial properties of PLLA as well as HAp are well-known. For example, the pristine poly(L-Lactide) do not inhibit S. aureus growth (Z.O. Erdohan, B. Cama, K.N. Turhan “Characterization of antimicrobial polylactic acid based films” Journal of Food Engineering 119 (2013) 308–315), L. monocytogenes, E. coli O157:H7, and S. Enteritidis (T. Jin, H. Zhang “Biodegradable Polylactic Acid Polymer with Nisin for Use in Antimicrobial Food Packaging“ Journal of Food Science 73(3) (2018), L. monocytogenes and S. typhimurium (J.-W. Rhim, S.-.I. Hong, C.-S. Ha “Tensile, water vapor barrier and antimicrobial properties of PLA/nanoclay composite films” LWT - Food Science and Technology 42 (2009) 612–617). The limited number of publication it is possible to find in the literature according to antimicrobial properties of pristine PLLA. Some modification of PLLA has been applied to obtained PLLA with antimicrobial properties (as adding olive leaf extract, nisin, etc). However we believe that the antimicrobial properties of PLLA can be improved also by incorporation of nanoapatite into PLLA matrix. It is known that pure nanoapatite possesses some antibacterial properties against P. aeruginosa (P. Sobierajska, A. Dorotkiewicz-Jach, K. Zawisza, J. Okal, T. Olszak, Z. Drulis-Kawa, R. J. Wiglusz “Preparation and antimicrobial activity of the porous hydroxyapatite nanoceramics” Journal of Alloys and Compounds 748 (2018) 179-187).

We do not agree that this manuscript contained only standard materials characterization. The novelty of our research are based on Eu3+-doped nanoapatite embedded into PLLA. The Eu3+ ions can be used as fluorescence label in bio-imaging and they allow to observe what is going on with an implant in time. For example, during biodegradation process of PLLA and HAp, the Eu3+ ions will release and luminescence intensity will decrease.

All the materials, chemicals and solvents along with their purity/grade and supplier should be listed under section 2.1 and not scattered in the manuscript.

Answer:

It has been corrected according to reviewer’s suggestion.

Various symbols were used for the degree sign in degC. Use the correct symbol throughout the manuscript.

Answer:

The symbols have been unified.

Section 2.4 describes the characterization of the materials. The authors should add the details for sample preparation for all methods described in this section.

Answer:

It has been completed.

The scope of introduction should be widen to appeal for a broader audience, for instance the versatile applicability of PLLA should be briefly demonstrated in a sentence by mentioning and giving examples for diverse fields, e.g. membrane supports (ACS Sustainable Chem. Eng., 2019, 7, 11885), oil adsorption (Appl. Sci. 2019, 9, 1014), food packaging (Materials, 2017, 10, 952), photocatalysis degradation (ACS Sustainable Chem. Eng., 2018, 6, 2445).

Answer:

The sentence about usage of PLLA in diverse fields has been added to the introduction.

The SEM images show the same features for all the materials shown in Figure 2. Consequently, only one representative SEM image will suffice in the manuscript. The rest can be moved to the supporting information.

Answer:

The representative SEM image has been chosen and the rest has been moved to the supporting information file.

In the caption of Figure 5 the panel designations should not be capital letters in order to be consistent with the rest of the manuscript.

Answer:

It has been corrected. Moreover, Figure 5 has been improved to be unify with all of the figures in the manuscript.

The 3 panels of Figure 9 can be merged into a single panel with 3 y-axis.

Answer:

It has been done.

The strain at break for PLLA has a significant error. The authors should elaborate on this and further explanation should be provided.

Answer:

In the paper there has been presented mechanical tests results for PLLA (Poly-L-lactide) composites based on the hydroxyapatites obtained by twin screw extrusion process. As far as it has been known, there is no flawless method of obtaining composite materials. Among three basic methods (polymerization in situ, solution-casting and melt mixing) there has been chosen melt mixing – extrusion in the twin screw extruder. This method has huge advantage providing a good filler (e.g. hydroxyapatite) distribution in polymer matrix. In our case, this is the most important aspect because there have been investigated the nanocomposites. There are another advantages related to this method like rapidity and continuity ect. On the other hand it should be taken into account that the PLLA is very sensitive material for degradation which occurs during extrusion process. In this case, the molecular weight of PLLA decreases and reduces the mechanical properties, especially tensile strength. It could be assumed that it is one of the most important reason. The effect of mechanical properties reduction is not new – it has been described earlier [1] for PLLA doped with carbon nanotube. In our case, taking into account the possible application for bone tissue engineering there has been decided to investigate and show the mechanical tests results to be fair and show also disadvantage of the method and material. Next reason for this effect is related to an adhesion or an interaction of polymer and filler. There has been used the calcium hydroxyapatite without any surface modification to avoid additional impurities, where adhesion is rather poor. It is well known that there are many adhesion promoters or coupling agents that would help in this specific case – we are also working in this area and hopefully we will be able to publish our results. Another reason is a universal Instron 5966 machine that was used to obtain the mechanical properties generating so high errors.

The “Discussion” section is numbered as section #1 but it should be #4 (line 306).

Answer:

This mistake has been corrected.

Overall it is unclear to the reader what the actual novelty and achievements are. The authors should stress what has already been known in the literature and what is the advancement of the field that was achieved in this work.

Answer:

The novelty of our research concerning embedding into PLLA matrix Eu3+- doped nanohydroxyapatite. Such materials possess additional function as possibility of diagnosis at the use of Eu3+ ions. We obtained composites aimed to theranostic application which possess such advantages as biocompatibility (PLLA, HAp), bioactivity, bone regeneration promotion (HAP) as well as bio-imaging possibility (Eu3+ions).

The conclusion section should have some quantitative statements summarizing the most important research outcomes of the manuscript.

Answer:

The sentence summarizing the most important outcomes has been added to the manuscript.

Bibliography

[1]          I. Armentano, D. Puglia, F. Luzi, C.R. Arciola, F. Morena, S. Martino, L. Torre, Nanocomposites based on biodegradable polymers, Materials (Basel). 11 (2018). doi:10.3390/ma11050795.

Round 2

Reviewer 1 Report

The revised version of the manuscript "The comprehensive approach to preparation and investigation of the Eu3+ doped hydroxyapatite/poly(L-lactide) nanocomposites: promising materials for theranostics application " by Wiglusz and coworkers shows a decent improvement.

All my comments were addressed. In my opinion Figure 1 from the reply letter should be included in the SI. thus, I suggest publication after minor revisions.

Author Response

Dear Editor,

We would like to express once again our sincerest gratitude to the Reviewers for their efforts put in criticizing the manuscript. We have taken into account all raised question here follows the detailed answers to the Reviewers. Moreover, all changes we have made to the original manuscript, are marked in the red color in the text.

Comments and Suggestions for Authors

The revised version of the manuscript "The comprehensive approach to preparation and investigation of the Eu3+ doped hydroxyapatite/poly(L-lactide) nanocomposites: promising materials for theranostics application " by Wiglusz and coworkers shows a decent improvement.

 All my comments were addressed. In my opinion Figure 1 from the reply letter should be included in the SI. thus, I suggest publication after minor revisions.

Answer:

The Figure 1 has been added into supplementary file as Figure S1. Moreover, the sentence: The particles size distribution based on SEM images of pure HAp and 3 mol% Eu3+:HAp particles as well as representative SEM images of powders were presented on Figure S1” has been added to the manuscript.

Reviewer 2 Report

The manuscript has considerable improved, the comments have been addressed. Most likely the authors used an automated reference manager as there are multiple mistakes in the reference list, which needs to be corrected.

Ref [4]: article number (035011) missing

Ref [8]: date should be 2019 instead of 2018, and 'Sustain.' should be 'Sustainable' ; add DOI as for the other references

Ref [9]: remove city name

Ref [10]: 'Sustain.' should be 'Sustainable'

Ref [14]: page numbers are missing (172-178)

Ref [32]: title and page nubmers are missing

Author Response

Dear Editor,

We would like to express once again our sincerest gratitude to the Reviewers for their efforts put in criticizing the manuscript. We have taken into account all raised question here follows the detailed answers to the Reviewers. Moreover, all changes we have made to the original manuscript, are marked in the red color in the text.

Comments and Suggestions for Authors

The manuscript has considerable improved, the comments have been addressed. Most likely the authors used an automated reference manager as there are multiple mistakes in the reference list, which needs to be corrected.

Ref [4]: article number (035011) missing

Ref [8]: date should be 2019 instead of 2018, and 'Sustain.' should be 'Sustainable' ; add DOI as for the other references

Ref [9]: remove city name

Ref [10]: 'Sustain.' should be 'Sustainable'

Ref [14]: page numbers are missing (172-178)

Ref [32]: title and page numbers are missing

Answer:

All mistakes in the references have been corrected.
